# Intranasal delivery of Thyroid hormones in MCT8 deficiency

Carmen Grijota-Martínez[1,2☯], Soledad Bárez-López[2,3☯], Eva Ausó[1,4], Samuel Refetoff[5,6,7], William H. Frey, II[8,9], Ana Guadaño-Ferraz[1,2]*

1 Instituto de Investigaciones Biomédicas Alberto Sols, Consejo Superior de Investigaciones Científicas (CSIC)-Universidad Autónoma de Madrid (UAM), Madrid, Spain, 2 Center for Biomedical Research on Rare Diseases (Ciberer), Unit 708, Instituto de Salud Carlos III, Madrid, Spain, 3 Translational Health Sciences, Bristol Medical School, University of Bristol, Bristol, United Kingdom, 4 Departamento de Óptica, Farmacología y Anatomía, Universidad de Alicante, Carretera de San Vicente del Raspeig s/n, Alicante, Spain, 5 Departments of Medicine, The University of Chicago, Chicago, Illinois, United States of America, 6 Departments of Pediatrics, The University of Chicago, Chicago, Illinois, United States of America, 7 Departments of Committee on Genetics, The University of Chicago, Chicago, Illinois, United States of America, 8 HealthPartners Neuroscience Center, St. Paul, Minnesota, United States of America, 9 HealthPartners Institute, St. Paul, Minnesota, United States of America

☯ These authors contributed equally to this work.
* aguadano@iib.uam.es

**Data Availability Statement:** All relevant data are within the paper.

**Funding:** This work was supported by the Spanish Ministry of Economy and Competitiveness, grant number SAF2017-86342-R (MINECO/AEI/FEDER,

## Abstract

Loss of function mutations in the gene encoding the thyroid hormone transporter monocarboxylate transporter 8 (MCT8) lead to severe neurodevelopmental defects in humans associated with a specific thyroid hormone phenotype manifesting high serum 3,5,3'-triiodothyronine (T3) and low thyroxine (T4) levels. Patients present a paradoxical state of peripheral hyperthyroidism and brain hypothyroidism, this last one most likely arising from impaired thyroid hormone transport across the brain barriers. The administration of thyroid hormones by delivery pathways that bypass the brain barriers, such as the intranasal delivery route, offers the possibility to improve the neurological defects of MCT8-deficient patients. In this study, the thyroid hormones T4 and T3 were administrated intranasally in different mouse models of MCT8 deficiency. We have found that, under the present formulation, intranasal administration of thyroid hormones does not increase the content of thyroid hormones in the brain and further raises the peripheral thyroid hormone levels. Our data suggests intranasal delivery of thyroid hormones is not a suitable therapeutic strategy for MCT8 deficiency, although alternative formulations could be considered in the future to improve the nose-to-brain transport.

## Introduction

Thyroid hormones (THs), thyroxine (T4) and 3,5,3'-triiodothyronine (T3), play an essential role in brain development and function [1]. Although the thyroid gland synthesizes mainly T4, most actions of THs are mediated by binding of the transcriptionally active form, T3, to its nuclear receptors regulating gene expression patterns [2]. In addition to genomic actions, both

UE) to AG-F, the Sherman Foundation (grant number OTR02211) to AG-F and SB-L, and the BBSRC (grant number BB/R016879/1) to SB-L. CG-M is a recipient of a contract from the Center for Biomedical Research on Rare Diseases (CIBERER), Instituto de Salud Carlos III, Madrid. S. R. was supported by grant DK15079 from the National Institutes of Health, USA. The cost of this publication has been paid in part by FEDER funds (European Funds for Regional Development). We acknowledge support of the publication fee by the CSIC Open Access Publication Support Initiative through its Unit of Information Resources for Research (URICI). The funders had no role in study design, data collection and analysis, decision to publish, or preparation of the manuscript.

**Competing interests:** The authors have declared that no competing interests exist.

T3 and T4 can also exert non-genomic actions through receptors in the plasma membrane, cytoplasm or mitochondria [3].

In the brain, T4 is the primary source of T3 [4, 5]. T4 crosses the brain barriers (BBs, including the blood–brain barrier, BBB, and the blood–cerebrospinal fluid barrier) through TH transporters into the astrocytes where it is converted to T3 by the action of the selenoenzyme deiodinase type II (DIO2) [6]. Although to a lesser extent, T3 can also be transported across the BBs directly into the extracellular fluid where it accesses the target neural cells [7]. Regarding TH transport across cellular membranes, among all the membrane proteins that can transport THs, the monocarboxylate transporter 8 (MCT8 [8]) and the organic anion transporter family member 1C1 (OATP1C1 [9]) are critical for the transport of THs across the BBs [10]. MCT8 shows high specificity for both T4 and T3 transport whereas OATP1C1 presents higher selectivity for T4. There are important differences between mice and humans regarding the expression of these transporter proteins in the brain. In both mice and humans, MCT8 is abundantly present in the endothelial cells of the BBB. In contrast, OATP1C1 is abundantly expressed in the mice BBB but is only present at very low levels in humans [11–14].

Inactivating mutations in the gene encoding for the MCT8 (*SLC16A2*) transporter located on the X chromosome [15] result in MCT8 deficiency, also known as the Allan–Herndon–Dudley syndrome, in males [16, 17]. Patients present abnormal TH concentrations in the serum with elevated T3 (leading to peripheral hyperthyroidism), low T4 and reverse T3 and normal or slightly elevated thyrotropin (TSH) [16–19]. Patients also suffer from severe neurological manifestations including global developmental delay, intellectual disability, lack of speech and motor impairments (central hypotonia with poor head control, progressive spastic quadriplegia and dystonic movements; [20, 21]). These alterations appear as a consequence of brain hypothyroidism [22–24], most probably caused by the lack of a functional MCT8 in the BBs impairing THs transport across the BBs to the brain [25–27].

Despite efforts to develop an effective therapeutic strategy [28], the treatments available so far have only been successful in ameliorating the peripheral hyperthyroidism [18, 29–31] but have failed to improve the neurological impairments. Since available evidence suggests that brain hypothyroidism in MCT8 deficiency arises from impaired TH transport across the BBs, the use of delivery pathways that bypass the BBs may offer opportunities for the treatment of the neurological defects of MCT8-deficient patients.

The nasal cavity provides a direct pathway to the brain that can be used to deliver therapeutic agents by intranasal administration bypassing the BBs widely demonstrated in preclinical and clinical studies [32, 33]. The intranasal delivery approach was developed by Frey [34], and it provides a non-invasive drug administration route that bypasses the BBs while reducing systemic exposure and therefore side effects [32]. Drugs administered in the nasal cavity can be transported directly to the brain along the olfactory [35] and trigeminal nerves [36, 37] as these neural pathways connect the nasal mucosa to the brain. This transport can take place through two different pathways. Using the intracellular pathway, drugs are internalized into vesicles by endocytosis in olfactory sensory neurons, they are then anterogradely transported along axons to the neuron's projection site where they are finally released by exocytosis [38]. In the extracellular pathway, drugs cross the nasal epithelium by paracellular transport, then travel along the perineural space until they reach the subarachnoid space of the brain [39] from which they rapidly travel to other brain regions via the perivascular spaces of the cerebrovasculature [40]. This delivery route has been successfully used to rapidly deliver and target insulin to the brain to improve memory in normal healthy adults [41] and in patients with Alzheimer's disease [42, 43] without altering the blood levels of insulin or glucose. Intranasal deferoxamine (656.8 g/mol), a high affinity iron chelator of a similar molecular size as T3

(650.97 g/mol), has been reported to target the brain to treat animal models of stroke [44], Alzheimer's [45] and Parkinson's [46].

The aim of this study was to explore the use of intranasal delivery as a potential route to administer THs to palliate the hypothyroid conditions of MCT8-deficient mice by promoting TH signaling in brain without worsening the peripheral hyperthyroidism. We found that, under the present formulation, intranasal administration of THs does not increase the content of THs in the brain and further raise the peripheral TH levels demonstrating that this is not a suitable therapeutic strategy for MCT8 deficiency. Alternative formulations, such as lipid emulsions or lipid nanoemulsions could be considered in the future to improve the nose-to-brain transport.

## Materials and methods

### Experimental animals

All experimental procedures involving animals were performed following the European Union Council guidelines (directive 2010/63/UE) and Spanish regulations (R.D. 53/2013), and were approved by the ethics committee Comité de Ética y Experimentacion Humana y Animal (CEEHA) at Consejo Superior de Investigaciones Científicas (CSIC) and by the Comunidad Autónoma de Madrid Review Board (proex 162/17) for the use of animals for scientific purposes. The present study was designed following the ARRIVE (Animal Research: Reporting of In Vivo Experiments) guidelines [47].

Mice were housed at the Instituto de Investigaciones Biomédicas "Alberto Sols" under temperature- and light-controlled conditions at 22°C on a 12:12 light–dark cycle (lights on at 7.00 am) with food and water *ad libitum*. Animals were housed in groups and were provided with environmental enrichment that included bedding and nesting material. Experiments were performed in Wild type (Wt), MCT8-deficient (*Mct8*KO) and double MCT8- and DIO2-deficient (*Mct8/Dio2*KO) mice at 3 months of age. Only males were used as MCT8 deficiency symptoms manifest in the hemizygous state. Wt and *Mct8*KO mice were originally generated by Dumitrescu and colleagues [22] and *Dio2*KO mice by Schneider and colleagues [48] in C57BL/6J genetic background and a colony was created at our animal facility in the same strain. The breeding strategy consisted on backcrossing $Mct8^{-/+}$ females with $Mct8^{+/y}$ males to produce Wt and *Mct8*KO littermates, and $Mct8^{+/y} Dio2^{-/-}$ males and $Mct8^{-/+} Dio2^{-/-}$ females to obtain *Mct8/Dio2*KO mice. The *Mct8* and *Dio2* genotypes were confirmed by PCR of tail DNA as described [49].

### Drug formulations

Drugs were prepared at concentrations suitable for intranasal administration. Since only 0.01–1% of an intranasally-administered drug is likely to reach the brain, a highly concentrated dosing solutions were used. L-T3 (Sigma, T6397) and L-T4 (Sigma, T2376) were dissolved in NaOH 0.04 N at 17.14 mg/ml and 24 mg/ml, respectively, and were further diluted into L-T3 0.214 mg/ml and L-T4 0.3 mg/ml in 0.9% NaCl (saline) on the day of the assay to a final pH = 8.

### Intranasal treatment

For intranasal administration mice where anaesthetized with isoflurane (4% for induction, 2% for maintenance in O2, Abbot Laboratories, Madrid, Spain). After induction, the animals were placed lying on their back and maintaining the head positioned in a way that prevents drainage

of the dose solution into the trachea and oesophagus. Eight 3-µl drops of the different TH solutions were administered intranasally every 2 minutes to alternating nares (a total of 24 µl).

Several treatments were performed. First, mice were treated either with saline or with a solution of 0.3 mg/ml L-T4 in saline: Wt + saline n = 3; Wt + T4 n = 3; *Mct8*KO + saline n = 2 and *Mct8*KO + T4 n = 5. In another experiment, mice were treated with saline or with a solution of 0.214 mg/ml L-T3 in saline: Wt + saline n = 7; Wt + T3 n = 7; *Mct8/Dio2*KO + saline n = 8 and *Mct8/Dio2*KO + T3 n = 8. Finally, L-T3 (0.214 mg/ml in saline) was intranasally-administered alone, bound to 1% bovine serum albumin (BSA) or bound to 1% BSA in the presence 1% of the vasoconstrictor phenylephrine hydrochloride (PHE; Sigma, P6126) to Wt animals: Wt + saline n = 11; Wt + T3 n = 11; Wt + T3-1% BSA n = 7 and Wt+T3-1% BSA-1% PHE n = 7. All animals were euthanized by transcardial perfusion under anesthesia 15 minutes after the last administration, as drug uptake in the cerebral cortex has been detected 5 minutes following intranasal administration [50]. Mice were anesthetized with ketamine (75 µg/g of BW) and medetomidine hydrochloride (1 µg/g of BW) and a sample of blood was extracted by retroorbital collection to be used for determination of T4 and T3 levels in plasma. Mice were then transcardially perfused with saline to remove blood from tissues and the cerebral hemi-cortices were collected for T4 and T3 determinations. In every case, individual mice were treated as the experimental unit.

## BSA-T3 binding assay

In order to evaluate the lowest concentration of BSA necessary to bind a solution of 0.214 mg/ml of L-T3, a BSA-T3 binding assay was performed. The ability of BSA to bind a L-T3 replacement dose of 0.020 mg/ml [51] was also assessed as a binding control.

Both L-T3 solutions were incubated for 20 minutes at room temperature (RT) with 18,000 cpm of [125]I-T3 and in presence of different concentrations of BSA: 0.25%, 0.5%, 1%, 1.5%, 2% and 5%. Prior to incubation [125]I-T3 was purified by removing contaminating iodide by electrophoresis. After incubation, samples were precipitated with 10% trichloroacetic acid by centrifuging 30 minutes at 2,800 rpm and the supernatants were removed. The [125]I-T3 bound to BSA precipitated was assessed by measuring the [125]I in a gamma counter detector. Percentage of binding was calculated by referring these measurements to standard samples solely containing the 18,000 cpm of [125]I-T3.

## Radioimmunoassays for T4 and T3 determinations in plasma and tissues

High specific activity [125]I-T3 and [125]I-T4 (3000 µCi/µg) were prepared by labelled with [125]I (Perkin Elmer, NEZ033A) (3–5)-T2 (Sigma, D0629) and L-T3 (Sigma, T2877), respectively. This was done according to Morreale de Escobar and colleagues [52] with the only difference that the separation of the radiolabelled products was performed by ascending paper chromatography in Butanol:Ethanol:Ammonia 0.5N (5:1:2) for 16 h. [125]I-T3 and [125]I-T4 were eluted in ethanol and kept at 4°C.

T3 and T4 were extracted as described [52]. In short, T3 and T4 were extracted from 80 µl of plasma samples with methanol (1:6), then evaporated to dryness and resuspended in the radioimmunoassay (RIA) buffer. In the hemi-cortices, T3 and T4 were extracted by using methanol-chloroform, back-extracted into an aqueous phase with 0.05% $CaCl_2$, and purified and eluted in 70% acetic acid using DOWEX AG 1-X2 columns (Bio Rad, 140–1251). Recoveries of extracted T3 and T4 were calculated by adding tracer amounts of [125]I-T3, [125]I-T4 to the initial homogenates. The purified extracts were used for T3 and T4 determinations by sensitive RIAs as previously described [52] with a range of 0.4–50 pg T3/tube and 2.5–320 pg T4/tube

and for both RIAs a 5% coefficient for intra-assay variation and a 10–15% coefficient for inter-assay variation.

## Statistics

Data are expressed as box and whisker plots with individual values. Chi-Square tests were performed to detect outliers, which were excluded from further analyses. Differences between means were obtained either by two-way analysis of variance (ANOVA) and Bonferroni's post hoc test to correct for multiple comparisons between groups that are split on two independent variables (with treatment and genotype as factors) or with one-way ANOVA and the Bonferroni's post hoc test to correct for multiple comparisons between unrelated groups. Significant differences are represented as *p<0.0332, **p<0.0021, ***p<0.0002 and ****p<0.0001. All analyses were performed using GraphPad Prism 8.

## Results

### Intranasal treatment with T4 increases plasma T4 levels

We did a pilot study to determine if THs significantly entered the blood following intranasal delivery, as this would complicate the assessment of direct nose-to-brain delivery. Treatment with T4 was chosen over that with T3, as T4 is the primary source of T3 in the brain and DIO2 activity can locally modulate the conversion of T4 into T3 to make it available to the appropriate target neural cells. For this reason, the MCT8-deficient animal model chosen for this set of experiments was *Mct8*KO mice, that has an elevated DIO2 activity in the brain [22, 23].

In order to assess the ability of T4 to reach the blood following intranasal administration, Wt and *Mct8*KO counterparts were treated intranasally with either saline or T4, and 15 minutes later, plasma samples were collected to assess the concentrations of T4. Intranasal treatment with T4 greatly increased plasma T4 levels in the following 15 minutes after delivery both in Wt (3.5-fold) and *Mct8*KO mice (5-fold; Fig 1) compared to controls given saline (Fig 1).

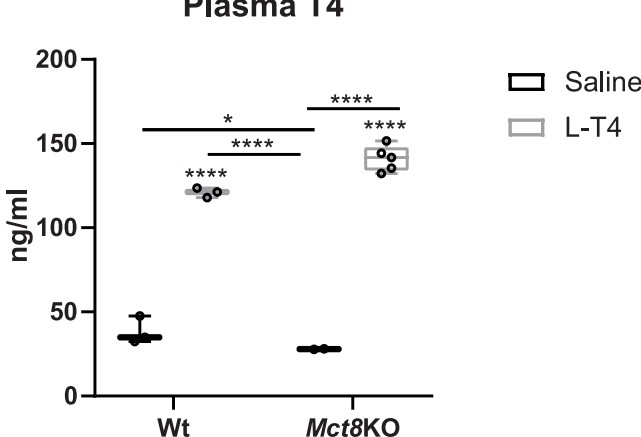

**Fig 1. Thyroxine (T4) levels in plasma of wild type (Wt) and MCT8 knockout (*Mct8*KO) mice 15 minutes after intranasal treatment with saline or L-T4.** Data are presented as box and whisker plots with individual values, showing median (horizontal line), boxes representing the 25th to 75th percentiles and whiskers representing the minimum and maximum values. P values *p<0.0332 and ****p<0.0001 were determined by two-way analysis of variance (ANOVA) and Bonferroni's post hoc test, the two factors being genotype and treatment. Significant differences with the saline-treated Wt group are indicated on top of each column, while significant differences between the remaining groups are indicated with lines.

Because T4 levels increase in plasma after intranasal delivery, and as mice abundantly present the T4 transporter OATP1C1 at the BBB, it would have not been possible to determine if a potential increase in T4 content in the cerebral cortex could be mediated by direct nose-to-brain transport or indirect access of T4 from the blood to the brain. In the direct nose-to-brain route, T4 administered transnasally would travel through extracellular pathways described for intranasal delivery from the nasal cavity to the brain. Alternatively, T4 accessing the blood stream following intranasal delivery could be being transported indirectly into the brain by crossing the BBB through the OATP1C1 transporter.

In view of the results obtained after intranasal administration of T4 to Wt and *Mct8*KO mice, it was essential 1) to establish if THs delivered intranasally can directly access the brain by nose-to-brain transport and 2) to reduce the amount of THs administered intranasally reaching the plasma in order to avoid worsening the peripheral hyperthyroidism characteristic of MCT8-deficient patients.

## Intranasal T3 treatment does not increase brain T3 content and further elevates plasma T3 levels

To determine if intranasal administration of THs can directly access the brain, we decided to execute an intranasal treatment with L-T3, which does not cross the BBB through OATP1C1, in double *Mct8/Dio2*KO mice, which present further reduced T3 content in the brain in comparison to single *Mct8*KO mice [53, 54] and therefore present higher sensitivity to detect increases in T3 brain content. For this, Wt and *Mct8/Dio2*KO mice were treated with either saline or L-T3 by intranasal administration and plasma and cerebral cortex samples were collected 15 minutes after for TH determinations.

In the saline-treated group, T3 and T4 levels in plasma reflected the characteristic thyroid function test in MCT8 deficiency with high T3 (3-fold increase) and low T4 (1.6-fold decrease) in *Mct8/Dio2*KO animals in comparison to Wt (Fig 2) [22, 23]. In the cerebral cortex, *Mct8/Dio2*KO mice displayed a 5-fold decrease in T3 content and a 1.6-fold decrease in T4 content, as previously described (Fig 2) [53, 54].

After intranasal treatment with L-T3, T4 levels in plasma and T4 content in the cerebral cortex were not affected either in Wt or in *Mct8/Dio2*KO mice. However, this treatment increased by 4-fold the plasma T3 levels in Wt animals and by 3-fold in *Mct8/Dio2*KO mice (Fig 2). In the cerebral cortex, intranasal treatment with L-T3 did not increase T3 content in *Mct8/Dio2*KO animals. However, in Wt animals there was a 1.3-fold increase in the content of T3 after intranasal L-T3 treatment, likely due to the presence of MCT8 at the BBs that can transport the increased levels of T3 in plasma to the brain.

## The use of carrier proteins or vasoconstrictors does not prevent T3 from reaching the blood

In order to inhibit THs from reaching the bloodstream following intranasal administration, the possibility of using carriers both suitable for intranasal administration and capable of binding THs was evaluated. A suitable potential carrier for TH intranasal administration is bovine serum albumin (BSA). Previous studies have demonstrated that BSA delivered intranasally is able to distribute throughout the mouse brain shortly after administration, with minimal penetration to the systemic circulation [50]. Moreover, BSA binds T3 with a relatively high affinity [55]. In view of this, we hypothesized that BSA can be used as a carrier to deliver T3 to the brain by means of intranasal administration reducing access to the bloodstream.

In order to determine the most suitable concentration of BSA that is able to bind a 0.214 mg/ml solution of L-T3, a radioactive T3-BSA binding assay was performed. A T3 replacement

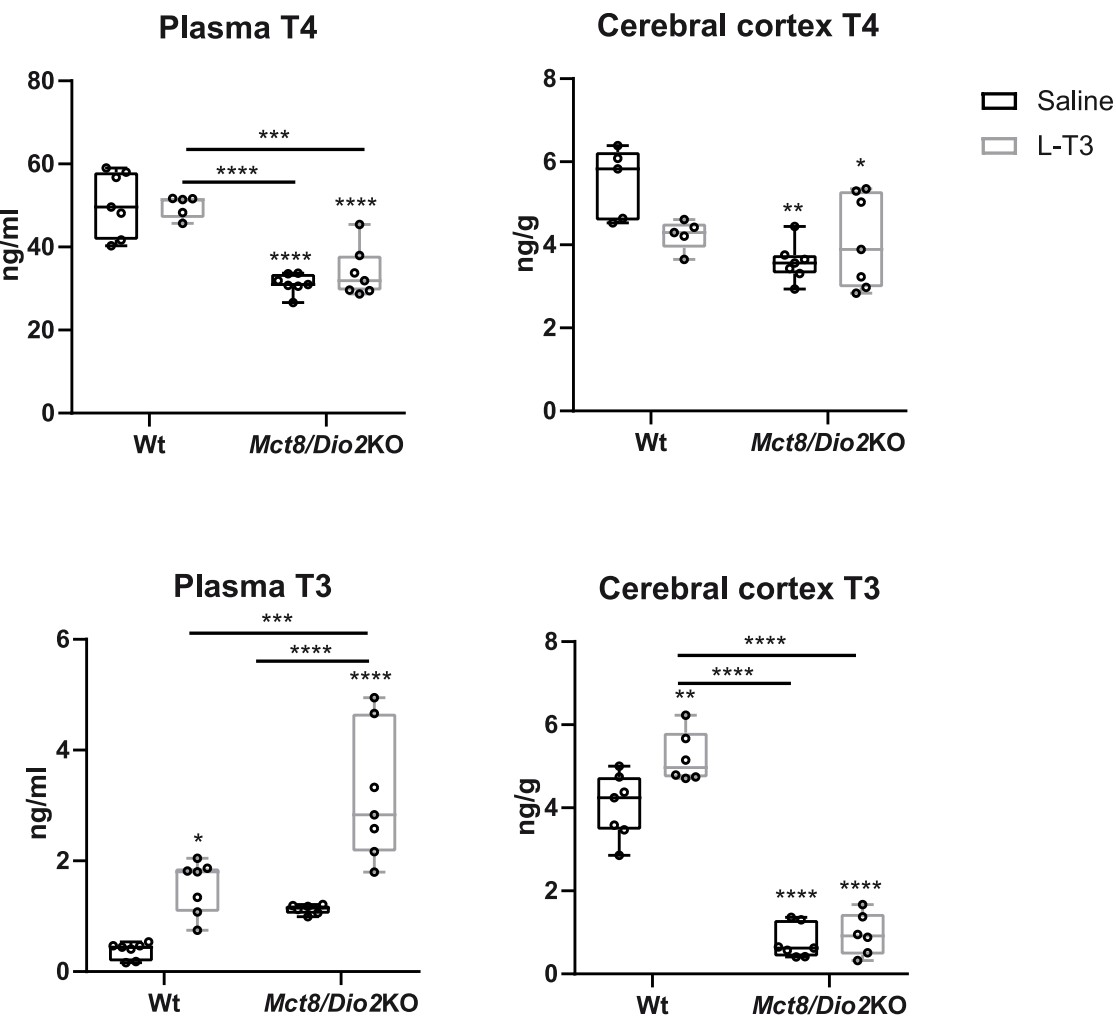

**Fig 2. T4 and 3,5,3'-triiodothyronine (T3) levels/content in plasma and cerebral cortex of Wt and MCT8 and DIO2 knockout (*Mct8/Dio2*KO) mice 15 minutes after intranasal treatment with saline or L-T3.** Data are presented as box and whisker plots with individual values, showing median (horizontal line), boxes representing the 25th to 75th percentiles and whiskers representing the minimum and maximum values. P values *p<0.0332, **p<0.0021, ***p<0.0002 and ****p<0.0001 were determined by two-way ANOVA and Bonferroni's post hoc test, the two factors being genotype and treatment. Significant differences with the saline-treated Wt group are indicated on top of each column, while significant differences between the remaining groups are indicated with lines.

dose of 0.020 mg/ml [51] was run in parallel as a binding control. Both L-T3 solutions were incubated with a tracer quantity of $^{125}$I-T3 in presence of different concentrations of BSA: 0.25%, 0.5%, 1%, 1.5%, 2% and 5%. As shown in Table 1, the ability of BSA to bind T3 was similar for both L-T3 solutions. 1% BSA was already binding 84.1% of the highly concentrated T3 and further concentrating BSA up to 5% only increased T3 binding in 6%. As Falcone,

**Table 1. Percentage of 0.214 mg/ml and 0.02 mg/ml solutions of L-T3 bound to 0.25%, 0.5%, 1%, 1.5%, 2% and 5% bovine serum albumin (BSA).**

|  | 0.25% BSA | 0.5% BSA | 1% BSA | 1.5% BSA | 2% BSA | 5% BSA |
|---|---|---|---|---|---|---|
| 0.214 mg/ml L-T3 | 70.5% | 79.7% | 84.1% | 87.8% | 88.06% | 90% |
| 0.020 mg/ml L-T3 | ND | 79.9% | 88.8% | ND | 94.7% | 97.2% |

ND = not determined.

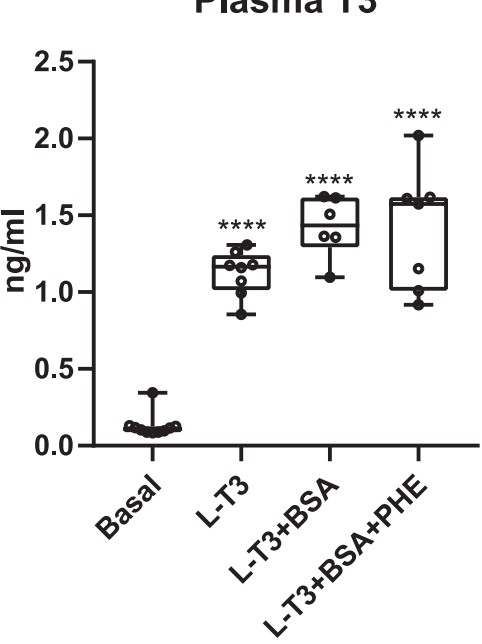

**Fig 3. T3 levels in plasma of Wt mice 15 minutes after intranasal treatment with L-T3, L- T3 bound to 1% bovine serum albumin (T3-BSA) or L-T3 bound to 1% BSA in the presence of 1% vasoconstrictor phenylephrine hydrochloride (T3-BSA-PHE).** Data are presented as box and whisker plots with individual values, showing median (horizontal line), boxes representing the 25th to 75th percentiles and whiskers representing the minimum and maximum values. P values ****$p < 0.0001$ was determined by one-way ANOVA and Bonferroni's post hoc test. Significant differences with the saline-treated Wt group are indicated on top of each column.

Salameh (50) used 1% BSA in their studies assessing the distribution of BSA throughout the brain and plasma after intranasal administration, and taking into account that 1% BSA binds 84.1% of a 0.214 mg/ml L-T3 solution, 1% BSA was considered the most suitable concentration to use as a carrier for intranasal administration of T3.

Other strategies that improve the delivery of drugs into the brain after intranasal administration yet decreasing the amount of drug absorbed into the systemic circulation consist on the use of vasoconstrictors. In particular, the vasoconstrictor phenylephrine (PHE) has been successfully used to increase the brain-to-plasma ratio [56].

Taking into account all the above, animals were treated either with L-T3, with L-T3 bound to 1% BSA or with L-T3 bound to 1% BSA and PHE 1%. As the aim of this experiment was to evaluate the capacity of BSA and PHE to prevent L-T3 from reaching the blood after intranasal administration, these treatments were performed only in Wt mice. Fifteen minutes after treatment, blood samples were collected to assess plasma T3 levels. T3 determinations in plasma revealed that, as expected, intranasal treatment with L-T3 reached the circulation and increased plasma T3 levels by 8.5-fold in comparison to untreated basal levels. Intranasal delivery of L-T3 bound to 1% BSA either in the absence or presence of the vasoconstrictor PHE was not able to prevent T3 from reaching the circulation as both treatments increased T3 levels in plasma more than 10-fold in comparison to untreated values (Fig 3).

## Discussion

Patients with inactivating mutations in the TH transporter MCT8 suffer from peripheral hyperthyroidism and brain hypothyroidism. The state of brain hypothyroidism leads to severe

neuropsychomotor abnormalities in all probability due to impaired transport of THs to the brain. No postnatal therapeutic strategies attempted so far have been able to improve the neurological defects. Since these alterations arise in part due to impaired TH transport across the BBs, a drug administration route that bypasses the BBs has the potential to deliver THs to the brain in the absence of MCT8. However, whether administration of THs to the brain during postnatal stages can revert the neurological impairments that have already taken place during early brain development [24] is an unresolved question. Some of the delivery routes that have the potential to circumvent the BBs include the intracerebroventricular and the intranasal administration pathways. Although both administration routes are routinely used for different treatments in humans, if successful, intranasal delivery presents certain advantages over intracerebroventricular administration, as the former is less invasive. In the present study, we aimed to evaluate the intranasal administration route as a pathway to deliver THs directly into brain in the absence of MCT8.

As T4 is the primary source of T3 in the brain, we treated *Mct8*KO mice with a 0.3 mg/ml solution of T4 by intranasal administration that led to increased plasma T4 levels. However, as *Mct8*KO present the T4 transporter OATP1C1 at the BBs, a potential T4 increase in the brain could be mediated by the transport of T4 from the blood to the brain through the BBB. In order to establish whether intranasal administration of THs can reach the brain by nose-to-brain transport, we treated *Mct8/Dio2*KO mice with a solution of 0.214 mg/ml L-T3, as the OATP1C1 transporter is not involved in T3 transport to the brain in mice [25]. Intranasal L-T3 administration further elevated plasma T3 levels in Wt and *Mct8/Dio2*KO mice and, most importantly, T3 content in the cerebral cortex of *Mct8/Dio2*KO animals did not increase after treatment while it did so in Wt animals. This can be explained by T3 transport from the plasma to the brain through the MCT8 transporter in Wt animals, especially since plasma T3 levels were elevated after treatment. It can be therefore concluded that intranasal treatment with THs in mice does not increase TH content in the brain and further increases plasma TH levels, aggravating the peripheral hyperthyroidism. It has been postulated that about 0.01 to 1% of a given drug reaches the brain after intranasal administration [57]. Based on this, considering that a total of 5 μg were delivered in the present study and taking into account the brain T3 content in basal conditions in each genotype (Fig 2), we estimate that this would have represented a- 1.25- to 25-fold increase in T3 brain content in a Wt animal and a 2- to 100-fold increase in T3 brain content in a *Mct8/Dio2*KO mouse. Therefore, an unsuitable dose of L-T3 is not likely to explain the lack of increase in T3 content after nasal treatment.

An additional potential drawback for intranasal TH treatment in MCT8-deficient patients identified in this study is the increase in the TH plasma levels following delivery. This would potentially worsen the peripheral hyperthyroidism with detrimental effects. There are some approaches that increase the brain-to-blood ratio and therefore reduce or even prevent the administered drug from reaching the blood. These include the use of carrier proteins and vasoconstrictors such as BSA and PHE, respectively. In previous studies, BSA reached the frontal, parietal and occipital cortex as early as 5 minutes and up to 4 hours after nasal delivery, and only 2–4% entered the systemic circulation [50]. Moreover, the vasoconstrictor PHE has been proven to reduce absorption of certain peptides into the blood by 65% after intranasal administration [56]. Based on these findings we treated Wt mice with L-T3, L-T3 bound to 1% BSA and L-T3 bound to 1% BSA in the presence of 1% PHE aiming to reduce the amount of T3 entering the systemic circulation. Unfortunately, neither of these treatments was successful in reducing or preventing T3 from reaching the bloodstream.

In the present study, there was a 1.18 ± 0.14 ng/ml increase in the plasma levels of T3 in Wt animals. Assuming a 1:1 ratio of blood cells to plasma [58], we estimate that 0.035% of the administered L-T3 entered the systemic circulation. We cannot explain why the use of BSA as

a carrier protein and the vasoconstrictor PHE was not able to reduce the amount of T3 reaching the bloodstream. The reason why some drugs reach the systemic circulation after intranasal delivery while others do not, is not fully understood [59]. It has been proposed that lipid soluble molecules can be rapidly absorbed from the nasal membrane into the bloodstream with a bioavailability of over 75%, while this percentage decreases to 10% for small, polar molecules and to less than 1% for large peptide molecules [60]. Despite the lipophilic nature of THs, it is very unlikely that these are being absorbed from the nasal epithelium to the blood after intranasal administration, as it has been established that THs need transporter proteins to cross cellular membranes [10]. Alternatively, THs could be being transported from the nasal epithelium to the blood by transporter proteins. In fact, the OATP3 transporter (encoded by *SLC21A7* and involved in TH transport) has been detected at remarkably high levels at the olfactory and nasal epithelia [61]. The presence of this transporter at the olfactory epithelium could be mediating a rapid transport of THs from the nose to blood, thus preventing THs from reaching the brain. Moreover, a quick degradation of THs at the nasal mucosa by DIO3 [62], that converts T4 and T3 into inactive metabolites, could also be responsible for the lack of TH increase in the brain. Due to the hydrophobic nature of THs and the solution used in this study, is it important to consider that THs could have precipitated in the nasal epithelium upon delivery. The possibility of using a lipid microemulsion formulation to intranasally administer TH to the brain could be considered for future approaches, as this has already been successfully used to deliver Growth differentiation factor 5, a water insoluble protein, to the brain [63].

Altogether, the findings of the present work indicate that the administration of THs by intranasal delivery is not suitable for the treatment of MCT8 deficiency, as this delivery route does not facilitate direct nose-to-brain transport of THs that would potentially ameliorate the brain hypothyroidism. Furthermore, this treatment increases TH plasma levels that would further aggravate peripheral hyperthyroidism characteristic of MCT8-deficient patients. However, our inability to formulate T3 and T4 in such a way as to successfully intranasally deliver and target them to the brain should not deter others from doing so, perhaps using a lipid emulsion or lipid nanoparticle formulation [64].

## Acknowledgments

We would like to thank Dr. Maria Jesus Obregon for valuable advice in thyroid hormone determinations and María Camino de Lucas for animal care.

## Author Contributions

**Conceptualization:** Carmen Grijota-Martínez, Soledad Bárez-López, Samuel Refetoff, William H. Frey, II, Ana Guadaño-Ferraz.

**Formal analysis:** Carmen Grijota-Martínez, Soledad Bárez-López.

**Funding acquisition:** Samuel Refetoff, Ana Guadaño-Ferraz.

**Investigation:** Carmen Grijota-Martínez, Soledad Bárez-López, Eva Ausó.

**Methodology:** Carmen Grijota-Martínez, Soledad Bárez-López, Eva Ausó, William H. Frey, II.

**Project administration:** Ana Guadaño-Ferraz.

**Supervision:** Ana Guadaño-Ferraz.

**Writing – original draft:** Soledad Bárez-López.

**Writing – review & editing:** Carmen Grijota-Martínez, Eva Ausó, Samuel Refetoff, William H. Frey, II, Ana Guadaño-Ferraz.

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
