## [Decision Letter · Decision Letter 0]

12 Jun 2020

PONE-D-20-16026

Intranasal delivery of Thyroid hormones in MCT8 deficiency

PLOS ONE

Dear Dr. Guadaño-Ferraz,

Thank you for submitting your manuscript to PLOS ONE. After careful consideration, we feel that it has merit but does not fully meet PLOS ONE’s publication criteria as it currently stands. Therefore, we invite you to submit a revised version of the manuscript that addresses the points raised during the review process.

Consequently, after considering the commentaries made by three reviewers, Major Revision is requested.

It is worth mentioning that such a decision is justified by taking into account the concerns raised by the reviewers regarding the statistical approach, as well as the publication requirements for PLOS ONE regarding that the papers must be technically sound and the data should support the conclusions.

Accordingly, please carefully address the issues raised in the comments provided by the reviewers appended below, which can help to improve and strengthen your submission.

We look forward to receiving your revised manuscript.

Kind regards,

Maria Carmen Iglesias-Osma, M.D., Ph.D.

Academic Editor

PLOS ONE

Journal Requirements:

2. At this time, we request that you  please report additional details in your Methods section regarding animal care, as per our editorial guidelines:

(1) Please state whether the provided ethics committee contains animal welfare experts or whether an animal ethics or IACUC committee reviewed and approved the study. Please provide the full name of the committee that reviewed and approved the study  

(2) Please include the method of euthanasia  

(3) Please provide the source of the animals used in the study.

Thank you for your attention to these requests.

Reviewers' comments:

Reviewer's Responses to Questions

**Comments to the Author**

1. Is the manuscript technically sound, and do the data support the conclusions?

Reviewer #1: Yes

Reviewer #2: No

Reviewer #3: Yes

2. Has the statistical analysis been performed appropriately and rigorously? 

Reviewer #1: Yes

Reviewer #2: No

Reviewer #3: Yes

3. Have the authors made all data underlying the findings in their manuscript fully available?

Reviewer #1: Yes

Reviewer #2: Yes

Reviewer #3: Yes

4. Is the manuscript presented in an intelligible fashion and written in standard English?

Reviewer #1: Yes

Reviewer #2: Yes

Reviewer #3: Yes

5. Review Comments to the Author

Reviewer #1: This manuscript reports a study of the effectiveness of intranasal delivery of thyroid hormone in the treatment of monocarboxylate transporter 8 (MCT8) deficiency. Loss of function mutations in MCT8 in humans lead to severe neurodevelopmental defects, apparently due to impairment in the transport of thyroid hormone across the blood brain barrier. Although these mutations result in hypothyroidism in the brain they produce hyperthyroidism in peripheral tissues. This manuscript describes a test of the ability of intranasal delivery of thyroid hormone as a means to bypass the brain blood barrier and selectively correct the hypothyroid neurological phenotype.

The authors tested intranasal delivery of T4 or T3 thyroid hormone in WT, MCT8 deficient, and double MCT8-/DIO2 deficient mice. Although the intranasal route greatly increased plasma T3 or T4 levels compared to controls, no, or only statistically non-significant increases in hormone in cerebral cortex in the MCT8 deficient animals were detected. Given the further increases in plasma thyroid hormone levels observed with intranasal administration were non-desirous, the authors also tested hormone formulations containing BSA (which binds T3 and T4) with or without phenylephrine (PHE) a vasoconstrictor. Neither formulation diminished the rise in serum thyroid hormone observed after intranasal administration. The authors conclude that their “data suggests intranasal delivery of thyroid hormones is not a suitable therapeutic strategy for MCT8 deficiency, although alternative formulations could be considered in the future to improve the nose-to-brain transport.”

The experiments described here appear to be well designed and carefully performed. Appropriate controls were employed, a variety of approaches were tested, the results are definitive, and the authors’ conclusions are convincing. Given that MCT8 deficiency in humans creates severe neurodevelopmental defects, the clinical need for exploring methods to ameliorate this defect is clear. Further, the authors’ approach to this goal was logical: intranasal delivery has been demonstrated previously as a non-invasive route that can, for several drugs, bypass the blood brain barrier. Although the ultimate results reported here were negative, i.e. that the intranasal route was not successful, this study is nonetheless valuable both for demonstrating that the intranasal approach as formulated here is not effective (important information for other scientists who might otherwise be tempted to try the same approach) and as a foundation for future approaches that may prove more useful. Very few specific concerns were noted:

1. All studies here assayed cerebral cortex levels of T3 or T4 thyroid hormone 15 minutes after administration. Why did the authors choose 15 mins. and has a more extensive time course been examined?

2. An interesting question is if counteracting the brain hypothyroidism in adults is likely to substantially ameliorate (at least in part) the neurological defects, or if the MCT8 defect may already have produced irreparable damage by the fetus/neonate stages. The authors should include a discussion of this issue.

3. The authors may wish to consider presenting their data as box-whisker plots, which would better visualize the statistics involved, rather than as columns; however this is not a requirement.

Reviewer #2: In the present manuscript, Grijota-Martínez et al. attempt to intranasally deliver thyroid hormones to the CNS to potentially treat centrally induced deficiencies associated with MCT8. My major concern is with the authors' conclusion that they are not able to increase thyroid hormones in the brain through the intranasal route. It seems that the experiments were not sufficiently powered to draw this conclusion. Data generated from the intranasal route is typically more variable than data generated by other routes of administration. The conclusion that intranasal administration of T4 does not increase brain levels based on an n=2 or 3 at a single 15 minute time point is not convincing. Two or three animals per treatment group is rarely sufficient to draw a statistically significant conclusion, especially with the intranasal route. The data shows a trend towards increased T4 in cortex after intranasal administration. This increase would likely be statistically significant if the study was not so underpowered. An n=6 animals per treatment group for the studies described in Figure 1 would be much more convincing. There is simply not enough data currently in the manuscript to conclude that intranasal delivery of thyroid hormones does not result in delivery to the CNS.

Reviewer #3: In this manuscript the authors describe studies where T4 and T3 were administered intranasally to mouse models lacking the thyroid hormone transporter MCT8 to determine if this could be an effective strategy to increase thyroid hormone (TH) levels in the brain. They also assessed and if this led to changes in circulating TH, and evaluated other strategies such as binding T3 with BSA or use of the vasoconstrictor phenylephrine to prevent TH spillover into the circulation. This study has important clinical applications since patients with MCT8 mutations have impaired TH transport across the blood brain barrier which leads to severe and devastating neurological impairment. Although the authors do not find that any of the protocols described are an effective strategy to alter TH levels in either of the MCT8KO models used, they present a strong starting point towards future studies which could lead to important therapies. These studies are clearly presented and the introduction and discussion are well written, informative, and balanced with the limitations considered and acknowledged.

1. In the discussion the authors elaborate on the rationale behind the dosages used but can they also discuss why they chose the timing that was used? Have they looked at TH content (or also possibly expression of TH-regulated genes) in the brain at later time points after administration?

2. In Fig. 2. why is the student’s t-test between saline treated WT and Mct8/Dio2KO also added in? The two-way ANOVA plus Bonferroni’s post test is more appropriate.

3. While these strategies may not have prevented spillover into the circulation they still may have facilitated uptake into the brain. Do the authors have any data in this regard?

4. Minor point: Figure 3 doesn’t have statistical differences indicated.

5. Lines 354-358. I don’t follow the logic of this section. If 0.01% to 1% of 5 micrograms of T3 was estimated to be delivered wouldn’t 0.5 to 50 ng to be delivered to both genotypes? Why are the estimated increases different between Wt and Mct8/Dio2KO? Please clarify.

6. PLOS authors have the option to publish the peer review history of their article (what does this mean?). If published, this will include your full peer review and any attached files.

Reviewer #1: No

Reviewer #2: No

Reviewer #3: No

---

## [Author Response · Author response to Decision Letter 0]

24 Jun 2020

The answer to reviewers and editor comments has been sent as an attach document

---

## [Editor Report · Decision Letter 1]

30 Jun 2020

Intranasal delivery of Thyroid hormones in MCT8 deficiency

PONE-D-20-16026R1

Dear Dr. Guadaño-Ferraz,

We’re pleased to inform you that your manuscript has been judged scientifically suitable for publication and will be formally accepted for publication once it meets all outstanding technical requirements.

Kind regards,

Maria Carmen Iglesias-Osma, M.D., Ph.D.

Academic Editor

PLOS ONE
---

## [Editor Report · Acceptance letter]

8 Jul 2020

PONE-D-20-16026R1 

Intranasal delivery of Thyroid hormones in MCT8 deficiency 

Dear Dr. Guadaño-Ferraz:

I'm pleased to inform you that your manuscript has been deemed suitable for publication in PLOS ONE. Congratulations! Your manuscript is now with our production department. 

Kind regards, 

on behalf of

Prof. Dr. Maria Carmen Iglesias-Osma 

Academic Editor

PLOS ONE